# Acidic pH Triggers Lipid Mixing Mediated by Lassa Virus GP

**DOI:** 10.3390/v12070716

**Published:** 2020-07-02

**Authors:** Uriel Bulow, Ramesh Govindan, James B. Munro

**Affiliations:** 1Department of Molecular Biology and Microbiology, Tufts University School of Medicine, Boston, MA 02111, USA; uriel.bulow@tufts.edu (U.B.); ramesh.govindan@tufts.edu (R.G.); 2Department of Microbiology and Physiological Systems, University of Massachusetts Medical School, Worcester, MA 01605, USA; 3Department of Biochemistry and Molecular Pharmacology, University of Massachusetts Medical School, Worcester, MA 01605, USA

**Keywords:** Lassa virus, virus entry, membrane fusion, single-particle fusion

## Abstract

Lassa virus (LASV) is the causative agent of Lassa hemorrhagic fever, a lethal disease endemic to Western Africa. LASV entry is mediated by the viral envelope glycoprotein (GP), a class I membrane fusogen and the sole viral surface antigen. Previous studies have identified components of the LASV entry pathway, including several cellular receptors and the requirement of endosomal acidification for infection. Here, we first demonstrate that incubation at a physiological temperature and pH consistent with the late endosome is sufficient to render pseudovirions, bearing LASV GP, non-infectious. Antibody binding indicates that this loss of infectivity is due to a conformational change in GP. Finally, we developed a single-particle fluorescence assay to directly visualize individual pseudovirions undergoing LASV GP-mediated lipid mixing with a supported planar bilayer. We report that exposure to endosomal pH at a physiologic temperature is sufficient to trigger GP-mediated lipid mixing. Furthermore, while a cellular receptor is not necessary to trigger lipid mixing, the presence of lysosomal-associated membrane protein 1 (LAMP1) increases the kinetics of lipid mixing at an endosomal pH. Furthermore, we find that LAMP1 permits robust lipid mixing under less acidic conditions than in its absence. These findings clarify our understanding of LASV GP-mediated fusion and the role of LAMP1 binding.

## 1. Introduction

Lassa virus (LASV) is an arenavirus endemic to Western Africa. An outbreak of Lassa fever in Nigeria led to 4841 suspected cases and 197 confirmed deaths in the first twenty-one weeks of 2020 [1]. The sequelae of Lassa fever include sensorineural deafness in 25% of survivors and a greater than 80% chance of maternal and/or fetal death in pregnant patients in their third trimester [2,3]. Due to the deadly nature of Lassa hemorrhagic fever, LASV is classified as a Category A priority pathogen by the United States Centers for Disease Control and Prevention.

The LASV envelope glycoprotein (GP) coordinates the fusion of the viral envelope with the endosomal membrane of the target cell [4]. GP is synthesized as a precursor glycoprotein complex that is subsequently processed by cellular proteases into subunits GP1 and GP2, as well as a long and stable signal peptide (SSP) [5,6]. During viral entry, GP1 is responsible for binding at least two identified cellular receptors, α-dystroglycan (αDG) and lysosomal-associated membrane protein 1 (LAMP1) [7,8]. Studies to date suggest a model in which GP1 first binds α-DG on the cell surface, which causes the cell to internalize the virus into an endosome via macropinocytosis [9]. The acidification of the late endosome causes GP to undergo an affinity switch from αDG to LAMP1 [10]. Implicated in this affinity switch are histidines H92, H93, and H230, which are located adjacent to the GP1 receptor-binding site and undergo conformational changes in response to protonation [11,12]. A crystallographic structure of GP at a neutral pH and cryo-electron tomography structures determined under acidic conditions have shown that acidic pH is sufficient to initiate conformational changes in GP1 [13,14].

The prevailing model of class I fusion holds that GP2 sits in a kinetically trapped, metastable pre-fusion conformation. The transition to the post-fusion conformation is facilitated by a trigger, which lowers the activation energy for exothermic collapse [15]. During fusion triggering, the GP2 subunit undergoes putative conformational rearrangements that involve the adoption of a transient extended intermediate in which the fusion loop is inserted into the target membrane [16]. The subsequent refolding of GP2 brings the target and viral membranes into close approximation, inducing fusion. Like other class I viral fusogens, the formation of the six-helix bundle is thought to drive membrane fusion [17,18,19].

Previous studies in LASV and lymphocytic choriomeningitis virus (LCMV) have highlighted the importance of acidic pH in arenavirus entry [14,20,21,22,23,24,25,26], with some cell–cell fusion studies suggesting that a pH as low as 3.5 may be required for LASV GP-mediated fusion to occur [27,28]. Of particular note, exposure to a pH consistent with the late endosome at a physiological temperature was shown to trigger the fusion of LCMV [29]. Here, we show that exposure to low pH is sufficient at a physiological temperature to trigger conformational changes in LASV GP related to membrane fusion. Furthermore, we show directly by a single-particle fusion assay that exposure to pH 5.0 is sufficient to trigger LASV GP-mediated lipid mixing in the absence of receptor binding. Finally, we demonstrate conclusively that LAMP1 binding is dispensable for LASV GP-mediated lipid mixing. However, its presence in the target membrane increases the kinetics of lipid mixing, facilitating robust lipid mixing at a less acidic pH.

## 2. Materials and Methods

### 2.1. Cell Culture

HEK293T cells were cultured in DMEM (Thermo Fisher, Waltham, MA, USA) supplemented with 10% cosmic calf serum (GE Healthcare, Chicago, IL, USA), l-glutamine (Thermo Fisher), and penicillin–streptomycin (Thermo Fisher) at 37 °C and 5% CO_2_ and were used to generate pseudovirions as described below. Vero cells were cultured in DMEM supplemented with 10% fetal bovine serum (FBS; Gemini Bio), l-glutamine, and penicillin–streptomycin at 37 °C and 5% CO_2_ and were used for infectivity assays with pseudovirions bearing GP and HA. DF1 chicken fibroblast cells were cultured in RPMI (Thermo Fisher) supplemented with 10% FBS, l-glutamine, and penicillin–streptomycin at 37 °C and 5% CO_2_, and were used for infectivity assays with pseudovirions bearing ALV env, as mammalian cells are neither susceptible nor permissive to infection by ALV subtype A.

### 2.2. Plasmid DNA

The pcDNA3 plasmid encoding LASV GP from the Josiah strain was provided by Dr. Melinda Brindley at the University of Georgia [11]. The pVRC8400 plasmid encoding IAV HA (A/Vietnam/2004) was provided by Dr. Peter Kwong at the Vaccine Research Center. HA0 was formed by site-directed mutagenesis as previously described [30]. ALV subtype A Env was provided by Dr. John Coffin at Tufts University. The soluble LAMP1 expression construct was provided by Dr. Juha Huiskonen at the University of Oxford [14].

### 2.3. Pseudotype Production

HEK293T cells were transfected with 10 μg of plasmid containing the glycoprotein of interest in a 10 cm dish at 80% confluency. PEI MAX was used for transfection at a mass ratio of 4:1 PEI to plasmid DNA. Cells were infected with VSVΔG-GFP-VSVG (Kerafast) 24 h post transfection at an MOI of 3 [31]. Supernatant was collected 24 h post infection and ultra-centrifugated over a 10% sucrose cushion at 160,000× *g* for 90 min. Supernatant was aspirated and pellets were resuspended in phosphate-buffered saline (PBS). The pseudovirus was aliquoted and frozen at −80 °C until use. Bald particles were generated following an identical protocol, with the exception of the omission of a glycoprotein.

### 2.4. Virus Infectivity Assays

Infections were performed in Vero cells for studies with LASV GP and IAV HA, and in DF-1 cells for ALV Env. 10 µL of concentrated pseudovirus were diluted in 0.1 M phosphate buffer (for pH 7.0 and 6.0) or 0.1 M acetate buffer (for pH 5.5 through 4.0) in temperature-controlled conditions. Virions were then brought back to neutral pH in 5× volume of 0.1 M phosphate buffer at pH 7.0 while still at their experimental temperatures, and then immediately serially diluted in room temperature DMEM for infection.

Target cells were infected with the pre-incubated virus and were incubated for one hour at 37 °C with rocking every 15 min. Fresh DMEM was then added to the cells. At 5 h post infection, cells were collected by trypsinization and assayed for GFP expression using flow cytometry on a FACSCALIBUR. Statistical significance was calculated using a two-way ANOVA with an alpha of 0.05.

### 2.5. LASV GP Antibody Capture ELISA

Purified VSV pseudotypes were treated with either a phosphate buffer at pH 6.0 or 7.0, or acetate buffer at pH 5.5 or 5.0 at 37 °C or on ice for 30 min. The virions were then brought back to a neutral pH, as described above, at their respective temperatures and bound to an ELISA plate. The plate was then blocked with a 3% BSA solution for 1 h and washed with PBS three times. The immobilized virions were then probed with non-neutralizing (26.5E, 24.6C) or neutralizing antibodies (12.1F, 37.2D, 37.7H) for 1 h, washed three times with PBS, and probed with an HRP-conjugated rabbit anti-human IgG1 antibody (Abcam, Cambridge, UK) [32]. The plate was developed with SureBlue™ TMB 1-Component Microwell Peroxidase Substrate (SeraCare, Milford, MA, USA) for twenty minutes and read in a Synergy H1 Biotek plate reader.

### 2.6. LAMP1 Purification

HEK293F cells were transfected with plasmid encoding the full lumenal domain of LAMP1 with a 6× His tag and grown for 6 days. PEI MAX was used as the transfection reagent as indicated above. Supernatant was collected and circulated over Ni-NTA agarose beads (Thermo Fisher, Waltham, MA, USA) overnight at 4 °C. The beads were then washed with PBS with 50 mM imidazole. LAMP1 was eluted from the beads with PBS with 250 mM imidazole. The protein was then dialyzed into PBS and stored at −80 °C in frozen aliquots.

### 2.7. Liposome Formation

Liposomes were formed as described in Floyd et al. [33]. Briefly, 1-oleoyl-2-palmitoyl-sn-glycero-3-phosphocholine (POPC) (Avanti Polar Lipids, Alabaster, AL, USA), 1,2,dioleoyl-sn-glycero-3-phosphocholine (DOPC) (Avanti Polar Lipids, Alabaster, AL, USA), cholesterol, 1,2-dioleoyl-sn-glycero-3-phosphoethanolamine-N-(carboxyfluorescein) (FLPE) (Avanti Polar Lipids, Alabaster, AL, USA), and either 1,2-dioleoyl-sn-glycero-3-[(N-(5-amino-1-carboxypentyl)iminodiacetic acid)succinyl] (Ni-NTA DOGS) (Avanti Polar Lipids, Alabaster, AL, USA) or bovine ganglioside GD1a (Millipore Sigma, Burlington, MA, USA) were mixed at a molar ratio of 4:4:2:0.2:0.1. Chloroform was evaporated under nitrogen gas, and lipids were resuspended in PBS to a working concentration of 10 mg/mL. The lipid mixture underwent seven freeze/thaw cycles in liquid nitrogen followed by extrusion through a 200 nm pore size polycarbonate membrane filter (Whatman, Maidstone, UK) 21 times.

### 2.8. Labeling Virions

Virions were labeled by incubating with 2 mM DiD (Thermo Fisher, Waltham, MA, USA) for two hours at room temperature. Unbound DiD was removed by passage over a Zeba desalting column (Thermo Fisher, Waltham, MA, USA) equilibrated in PBS. Labeled virus was used immediately in single-particle lipid mixing assays.

### 2.9. Single-Virion Lipid Mixing Assay

The single-virion lipid mixing assay was adapted from that described by Floyd et al. [33]. Pre-drilled quartz slides and glass coverslips were immersed in distilled water and sonicated for 20 min. They were then moved to 1 M KOH and the sonication was repeated. The slides were rinsed with distilled water, before being sonicated in acetone, and rinsed again. The coverslips and slides were then allowed to dry overnight. The dry slides were then cleaned in an plasma cleaner (Harrick Plasma, Ithaca, NY, USA) for 2 min. Flow cells were then constructed with the slides and coverslips. Each flow cell was rinsed with PBS, and then 100 μL of liposomes were injected into the flow cell by a syringe pump at 80 μL/min. The liposomes were allowed to incubate for twenty minutes at room temperature before the channel was again rinsed with 300 μL of PBS. Labeled virions were then pumped into the channel at 40 μL/min and allowed to incubate for 20 min to allow virions to bind to the membrane. The channel was then rinsed with PBS again to remove unbound virus. The slide was then placed on the stage of a prism-based TIRF microscope. Pre-warmed buffer was pumped into the flow cell while acquiring movies at 100 ms exposures (10 Hz) for a minimum of 3 min, or until no more dequenching events could be seen. The analysis of the dequenching events was performed in Matlab (MathWorks, Waltham, MA, USA). The interval between 50% loss of CF fluorescence resulting from acidification and DiD dequenching was extracted for each event and compiled into histograms. The histograms displayed a rise and fall in the frequency of lipid mixing events. This indicates that multiple rate-limiting steps are necessary before the arrival of lipid mixing; a single rate-limiting step would have given rise to an exponentially distributed histogram of the frequency of events. We therefore considered the kinetic scheme
$$A\to I_{1}\to I_{2}\to \cdots \to I_{N}\to B$$
where *A* represents the pre-fusion state, *B* represents the state after lipid mixing, and *I*_1_, *I*_2_,…*I_N_* are the *N* rate-limiting intermediate states. The probability density function for this scheme, which describes the distribution of time intervals observed for the transit from *A* to *B*, is the gamma distribution
$$P\left(t\right)=\frac{k^{N}t^{N-1}}{\Gamma \left(N\right)}e^{-kt}$$
where *k* is the rate constant and *N* is the number of rate-determining steps. The histograms of the time intervals were fit to the gamma distribution using a least squares algorithm in Matlab. The mean time to dequenching was then determined by calculating ⟨t⟩=N/k.

## 3. Results

### 3.1. The pH Sensitivity of LASV GP

We first sought to determine the pH sensitivity of LASV GP with a viral inactivation assay, a common strategy used to study the pH dependence of viral fusogen function [34]. We therefore formed GFP-encoding vesicular stomatitis virus (VSV) pseudovirions bearing GP from the Josiah strain of LASV. This pseudotyping strategy has previously been used for mutagenic studies of LASV GP, as well as for LASV vaccine trials [11,35]. VSV-GP pseudovirions were first incubated at 21 °C for five or thirty minutes in buffer at pH values ranging from 5.0 to 7.0. The virions were returned to a neutral pH while maintaining the temperature before they were used to infect Vero cells. GFP expression was assayed using flow cytometry five hours post infection. Above pH 5.0, no loss of infectivity was seen after five minutes of incubation at 21 °C (Figure 1A). Infectivity was reduced by approximately 13% when the virions were incubated at pH 5.0 for five minutes. A similar loss of infectivity was seen for pseudovirions carrying H5 hemagglutinin (HA) from influenza, which is known to be triggered by acidic pH [34,36]. When the virions were incubated at 37 °C and pH 5.5 for five minutes, infectivity was reduced to 45%. Infectivity was further reduced by 90% when incubated at pH 5.0 (Figure 1A). When the incubation was extended for thirty minutes, infectivity was reduced by 90% at pH 5.5 (Figure 1B). Again, similar losses of infectivity were seen for particles carrying HA. In contrast, no loss of infectivity was seen for pseudovirions carrying the envelope glycoprotein from avian leukosis virus (ALV Env) following incubation for five or thirty minutes at pH 5.0 at all temperatures tested (Figure 1). Finally, consistent with the model in which GP is kinetically trapped in the metastable pre-fusion conformation, no loss of infectivity was observed at any pH while on ice (Figure 1). These data demonstrate that incubation at pH 5.0 to 5.5 leads to loss of infectivity of GP-bearing pseudovirions at a physiologic temperature.

### 3.2. Acidic pH Triggers Loss of Recognition by Neutralizing Antibodies

To further test whether exposure to acidic pH at a physiological temperature triggers a conformational change in LASV GP, we evaluated the recognition by neutralizing antibodies 12.1F, 37.2D, and 37.7H using an enzyme-linked immunosorbent assay (ELISA). These antibodies recognize epitopes that bridge both GP1 and GP2, making them highly specific for the pre-fusion conformation of GP [13,32]. Virions were incubated at pH values ranging from 5.0 to 7.0 on ice or at 37 °C for thirty minutes. The pH was then neutralized while maintaining temperature, and virions were subsequently bound to an ELISA plate at room temperature. We then probed for the presence of the pre-fusion conformation of GP with neutralizing antibodies 12.1F, 37.2D, and 37.7H. Irrespective of temperature, exposure to pH 6 or 7 led to no loss of antibody binding (Figure 2). At pH 5.5, moderate loss of binding by only 37.7H was seen after a 37 °C incubation, whereas full binding was maintained after incubation on ice. Incubation at pH 5.0 at 37 °C abrogated binding by approximately 50% in the case of 12.1F, by more than 60% in the case of 37.2D, and by 75% in the case of 37.7H, whereas no loss of binding of any antibody was seen after incubation on ice. In contrast, no loss of binding after incubation at either temperature across the range of pH values was seen for the non-neutralizing antibodies 26.5E and 24.6C (Figure 2) [32]. Taken together, the infectivity and antibody binding data suggest that GP undergoes a pH-induced transition out of the pre-fusion conformation, concurrent with a loss of function. Here again, only when incubations were performed at a physiological temperature did the putative transition occur at a pH consistent with the late endosome.

### 3.3. Acidic pH Triggers Lipid Mixing Mediated by LASV GP

Our observations thus far support a hypothesis in which acidic pH is sufficient to promote the transition of GP from the pre-fusion to the post-fusion conformation. We sought to test this hypothesis by directly visualizing the pH-induced triggering of GP-mediated membrane fusion. To this end, we developed a single-particle lipid mixing assay adapted from those used in studies of influenza [33,37,38], West Nile virus [39], feline coronavirus [40], and VSV [41]. We formed a planar lipid bilayer supported within a quartz microfluidic cell. Contained within the bilayer was lipid-soluble 6-carboxyfluorescein (CF), a pH-sensitive dye used as a fluorescent pH indicator. VSV-GP pseudovirions were generated as described above, but with the additional inclusion of HA with a mutated furin cleavage site (HA0). This HA0 mutant is incapable of mediating fusion but enabled the attachment of the virions to the lipid bilayer by way of binding sialic acid, which was included in the bilayer in the form of the GD1a ganglioside. This approach allowed us to consider the role of pH in GP-mediated lipid mixing independently from the engagement of a receptor in the target membrane. The viral membrane was labeled with the lipophilic fluorophore DiD at a concentration sufficient to partially quench fluorescence. Based on the previous applications of this approach, we anticipated that the hemifusion of the viral membrane and the planar bilayer would allow the diffusion of the DiD into the bilayer, which would give rise to a sharp increase in fluorescence as the DiD dequenches. The DiD fluorescence then decays as the dye continues to diffuse into the planar bilayer. We bound VSV-HA0-GP virions to the supported lipid bilayer in the microfluidic channel at pH 7.0 and imaged the surface using prism-based total internal reflection fluorescence (TIRF) microscopy (Figure 3A). Puncta of DiD fluorescence indicated the presence of virions attached to the surface. Pre-warmed buffers at various pH values were then introduced into the flow cell with a syringe pump with the continuous monitoring of fluorescence.

The introduction of pH 4.0 buffer pre-warmed to 37 °C induced a precipitous loss of CF fluorescence, indicating the successful delivery of low pH buffer to the flow cell. Following the acidification of the chamber, the DiD puncta exhibited the characteristic sharp increase and trailing loss of fluorescence indicative of successful lipid mixing (Figure 3B, Appendix A). The integration of the fluorescence intensity of each punctum allowed for the generation of CF and DiD fluorescence traces (Figure 3C). Pseudovirions that had been incubated at acidic pH at 37 °C prior to binding to the planar bilayer displayed no evidence of lipid mixing upon the introduction of acidic buffer to the flow cell. Similarly, virions lacking GP and possessing only HA0 bound the surface but also did not display any evidence of lipid mixing, indicating that the observed lipid mixing was mediated by GP (Appendix A).

To determine the kinetics of lipid mixing, we extracted the time interval between the drop in CF fluorescence and the dequenching of DiD fluorescence. These intervals were compiled into histograms and fit to a gamma distribution, which allowed the determination of the rate constant, the number of rate-determining steps leading to lipid mixing, and the average time to lipid mixing (Figure 4A, Section Materials and Methods) [33,37]. This fitting procedure indicated that at pH 4.0, there existed approximately 4.2 ± 0.1 rate-determining steps leading to lipid mixing, with a rate constant of 0.31 ± 0.01 s^−1^, which yielded an average time to lipid mixing of 13.5 ± 0.5 s. Increasing the pH to 4.5 and 5.0 slowed the rate constant to 0.25 ± 0.01 and 0.19 ± 0.02, respectively, which led to corresponding increases in the time to lipid mixing (Figure 4B,C, Appendix A). However, across all pH values considered, the number of rate-determining steps between acidification and lipid mixing was consistently approximately 4 (Figure 4B). No lipid mixing was seen above pH 5.0 on the timescale of our movies. Below pH 4.0, lipid mixing occurred too rapidly to accurately evaluate in this assay. In summary, the timing of lipid mixing occurred in a pH-dependent manner, with the kinetics decreasing with increasing pH. The number of rate-determining steps was not sensitive to pH.

### 3.4. LAMP1 Increases the Kinetics of Lipid Mixing

We next asked whether the attachment of the virion to the target membrane by way of GP–LAMP1 interactions would affect the kinetics of lipid mixing. To this end, the single-virion lipid mixing experiment was repeated using VSV-GP pseudovirions lacking HA0, and with a soluble form of LAMP1 immobilized on the planar bilayer by way of a polyhistidine (6× His) tag bound to a Ni-NTA lipid. Virions were bound to the supported bilayer at pH 5.5. As in the absence of LAMP1, pH 5.5 was not sufficiently acidic to trigger lipid mixing. However, rapid lipid mixing was observed at pH 5.0 in the presence of LAMP1 (Appendix A). Kinetic analysis indicated a rate constant and average time to lipid mixing consistent with that observed at pH 4.0 in the absence of LAMP1 (Figure 4). The number of rate-determining steps (4.1 ± 0.1) remained similar to that seen in the absence of LAMP1 (Figure 4B). Thus, the interaction of GP with LAMP1 increased the kinetics of lipid mixing to an extent equivalent to reducing the pH by approximately one unit. Here again, pseudovirions that had been pre-incubated at an acidic pH at 37 °C prior to binding to the planar bilayer in the presence of LAMP1 displayed no sign of lipid mixing after the introduction of acidic buffer to the flow cell.

## 4. Discussion

Here, we demonstrate that acidic pH alone is sufficient to induce the conversion of LASV GP to a conformation that is unable to support entry into cells and unable to bind neutralizing antibodies that are specific to the pre-fusion conformation of GP. Importantly, only at a physiological temperature is pH 5.0, which approximates the pH of the most acidic endosomes [42], sufficiently acidic to promote this transition. No loss of infectivity or loss of antibody binding was seen when virions were incubated at room temperature or on ice at pH 5.0. This temperature-dependent transition to an irreversible conformation is consistent with a kinetically trapped pre-fusion conformation of GP. This implies that a large activation energy limits spontaneous transition out of the pre-fusion conformation into a conformation with much lower free energy. Elevated temperature increases the probability of spontaneously crossing this high activation energy, while the protonation of key residues in GP by the acidic environment likely reduces the activation energy. Thus, reduced temperatures, which lower the probability of crossing the activation energy, require more acidic pH to trigger conformational changes in GP [43]. This underscores the importance of maintaining a physiological temperature when characterizing the activity of glycoproteins such as LASV GP.

By attaching virions to a target membrane by orthogonal means—through HA0-mediated binding to sialic acid—we have further provided direct evidence that acidic pH is sufficient to trigger LASV GP-mediated lipid mixing in the absence of interaction with either ɑDG or LAMP1. Thus, no receptor-binding event is strictly necessary to trigger GP-mediated lipid mixing. These data provide support for the previous observation that the disruption of the ɑDG binding site through mutagenesis only minimally affects cell–cell fusion [11]. These data also support previous studies indicating that LAMP1 is not necessary for GP-mediated cell–cell fusion [21], nor is it strictly required for entry into cells [20].

Our kinetic analysis of single-virion lipid mixing events indicates that the time from pH drop to lipid mixing decreases at more acidic pH. This provides direct evidence for the pH-induced reduction in the activation energy that kinetically traps GP in the pre-fusion conformation. Interestingly, although the rate of lipid mixing increased with decreasing pH, the number of rate-determining steps remained approximately constant across the pH range tested. This indicates that the mechanism of GP-mediated lipid mixing is consistent across a range of pH values. This may be an indirect observation of intermediate conformations in GP, which may resemble those postulated for other class I viral fusogens, such as influenza HA and Ebola GP [16,30,43,44]. A cryo-electron tomography study of GP documented a pH-induced transition of GP to a conformation distinct from the pre- and post-fusion conformations [14], which may reflect an intermediate conformation achieved during fusion. Alternatively, the observation of multiple rate-determining steps leading to lipid mixing may be an indication that multiple GP trimers need to engage the target membrane in order to promote lipid mixing.

Finally, we have directly shown that the attachment of virions to the target membrane by way of binding LAMP1 increases the kinetics of lipid mixing. When LAMP1 is present, lipid mixing occurs at pH 5.0 at a rate approximately equal to the rate observed in the absence of LAMP1 at pH 4.0. Thus, while LAMP1 is not necessary for GP to promote lipid mixing, LAMP1 increases the pH at which GP-mediated lipid mixing occurs, thus facilitating LASV entry under less harsh conditions, as previously suggested [20]. This indicates that LAMP1 and acidic pH synergistically reduce the activation energy that limits GP conformational changes that lead to fusion. This observation is entirely consistent with previous studies demonstrating that LAMP1 increased the pH at which GP could mediate cell–cell fusion [21], and that virions enter by way of less acidic endosomes in cells expressing LAMP1 as compared to cells lacking LAMP1 [20]. Taken together, the present and previous studies provide a consistent understanding for the roles that endosomal pH and LAMP1 play in LASV entry. Future studies will be aimed at probing the mechanism by which acidic pH and LAMP1 enable GP conformational changes. This will likely require a combination of structure determination and biophysical interrogations equipped to visualize GP conformational changes that lead to fusion.

## Figures and Tables

**Figure 1 viruses-12-00716-f001:**
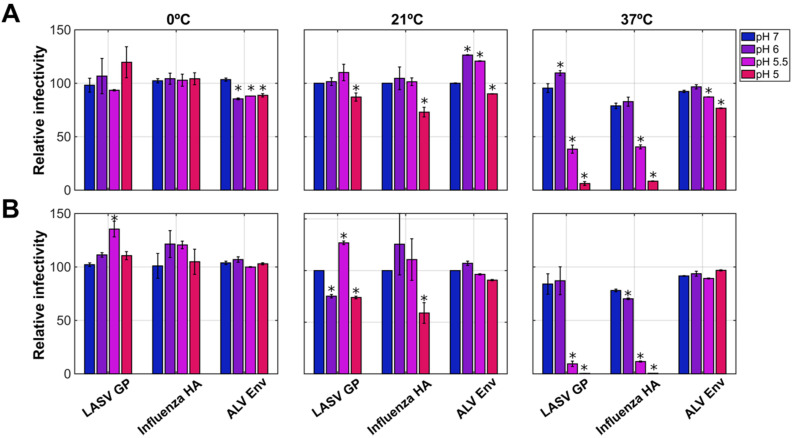
Exposure to low pH at physiological temperature inactivates LASV GP. VSVΔG-GFP-GP viral pseudotypes were treated with either phosphate buffer at pH 6.0 or 7.0, or acetate buffer at pH 5.5 or 5.0 at the indicated temperature for (**A**) 5 min or (**B**) 30 min. Virus was subsequently brought back to neutral pH and used to infect Vero cells. Infectivity was determined by measuring GFP expression using flow cytometry. Infectivity was normalized to the percent of GFP-positive cells at the 21 °C pH 7.0 condition for each virus. Assays were performed in biological triplicate with new viral preps, and each biological replicate was performed in technical triplicate. Data are presented as the mean of three biological replicates and error bars represent the standard error of the mean. An asterisk indicates a statistically significant difference as compared to the case of pH 7 (*p* < 0.05) calculated using a one-way ANOVA. The lack of an asterisk indicates no statistically significant difference as compared to pH 7.

**Figure 2 viruses-12-00716-f002:**
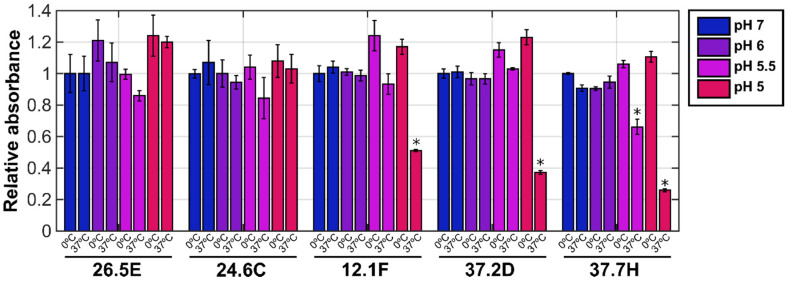
The binding of neutralizing and non-neutralizing antibodies to LASV GP after exposure to acidic pH at physiological temperature. VSVΔG-GFP-GP viral pseudotypes were treated with either phosphate buffer at pH 6.0 or 7.0, or acetate buffer at pH 5.5 or 5.0 at the indicated temperature for 30 min. Virus was subsequently brought back to neutral pH and assayed for binding by non-neutralizing (26.5E, 24.6C) and neutralizing antibodies (12.1F, 37.2D, 37.7H) by ELISA [32]. Data are presented as the mean of three biological replicates and error bars represent standard error of the mean. An asterisk indicates a statistically significant difference as compared to the case of incubation at 0 °C and pH 7 (*p* < 0.05.) calculated using a one-way ANOVA. The lack of an asterisk indicates no statistically significant difference.

**Figure 3 viruses-12-00716-f003:**
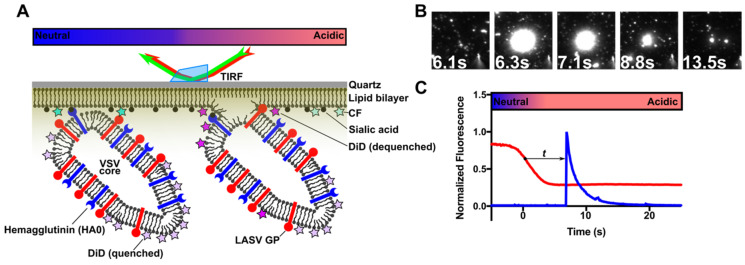
LASV GP mediates lipid mixing between fluorescently labeled virions and a supported lipid bilayer. (**A**) Cartoon of pseudovirions formed with a VSV core, inactive HA0, and LASV GP, hemifusing to a planar lipid bilayer supported by a quartz microscope slide within a microfluidic chamber. Pseudovirions were flowed onto the bilayer and allowed to attach at neutral pH by way of the HA0–sialic acid interaction. DiD fluorescence dequenching arose due to single pseudovirions hemifusing to the supported bilayer upon the introduction of acidic pH. CF was included in the supported bilayer in order to provide a fluorescence indicator of pH. Fluorescence was detected using the evanescent field generated from the simultaneous total internal reflection of 642-nm and 532-nm lasers on a custom-built prism-based TIRF microscope. Movies were recorded at a 100 ms exposure time (10 Hz) for a minimum of 1400 frames. (**B**) Frames from a single DiD dequenching event at pH 5 acquired at the indicated time points, where *Time = 0* was defined as a 50% decrease in CF fluorescence. (**C**) Representative fluorescence traces (CF, red; DiD, blue) are from a single dequenching event. The drop in CF fluorescence arises from the acidification of the buffer in the channel. The subsequent spike in DiD fluorescence arises from dequenching as DiD diffuses into the planar bilayer. The interval (*t*) between the 50% decrease in CF fluorescence and DiD dequenching was extracted for kinetic analysis.

**Figure 4 viruses-12-00716-f004:**
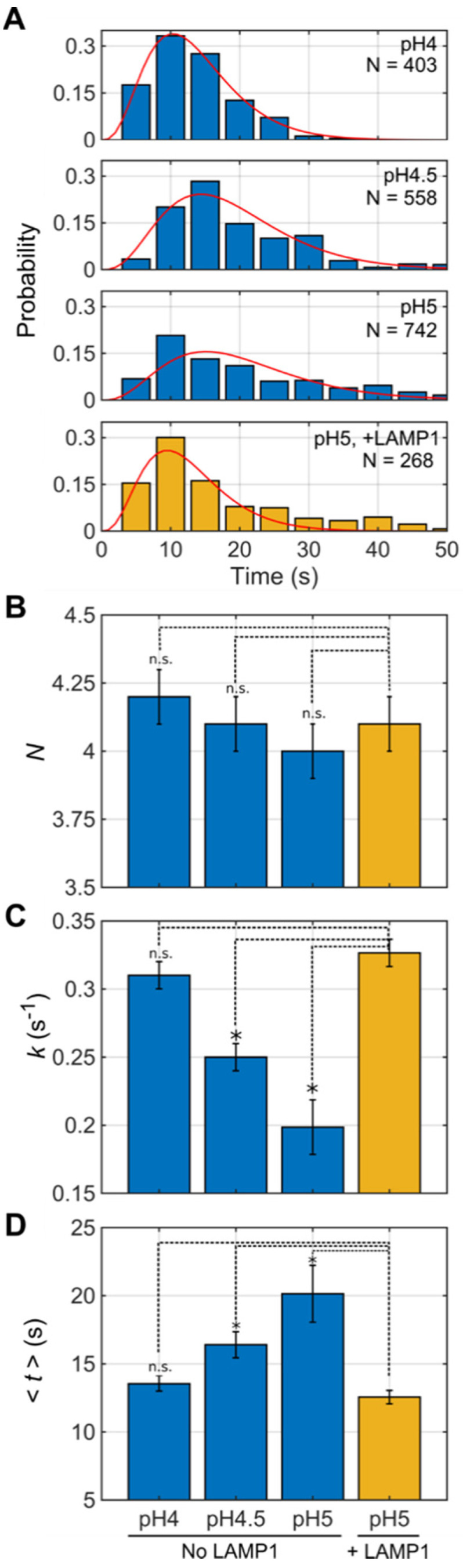
LAMP1 increases the kinetics of GP-mediated lipid mixing. For each lipid mixing event, the interval of time between the 50% decrease in CF fluorescence and DiD dequenching was compiled into a histogram and fit to a gamma distribution by a least-squares algorithm. (**A**) The histogram for each pH considered, in the absence or presence of LAMP1, is shown with the gamma distribution fit overlaid in red. *N* indicates the number of individual lipid mixing events. (**B**) For each experiment, the number of rate-determining steps leading from the acidification of the flow cell to dequenching was determined through the gamma distribution fitting of each histogram in A. (**C**) The rate constants (*k*) determined from the gamma distribution fitting of each histogram in A. (**D**) The mean time to dequenching, defined as *N/k* for a gamma distribution. Error bars reflect the 95% confidence intervals determined in the fitting. An asterisk indicates a statistically significant difference as compared to the case of LAMP1 at pH 5 (*p* < 0.05.) calculated using a one-way ANOVA.

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
