# Peer review of "Acidic pH Triggers Lipid Mixing Mediated by Lassa Virus GP"

_viruses, 2020, doi:10.3390/v12070716_

Round 1
Reviewer 1 Report
Lassa virus (LASV) surface glycoprotein (GP) possesses receptor binding and fusogenic properties which are essential for the delivery of the virus genome with associated proteins into the interior of the infected target cells. It has been shown that the initial steps of LASV infection requires a receptor exchange after virus uptake into the cellular vesicles and thereafter endosomal acidification for fusion of virus membrane with the endosomal membrane. Previous reports demonstrated that this fusion process only occurs at extreme low pH (< 4), which is a non-physiological role and persisted as an unsolved enigma for several years. The authors addressed the untackled problem by using a commercial pseudovirus system based on VSV expressing the LASV glycoprotein. They demonstrated that low pH values lead to loss of virus infectivity. Infectivity and antibody binding studies at pH at 37°C or 0°C and descending pH values suggested that the GP undergoes a pH-induced transition out of the pre-fusion conformation, concurrent with the loss of function. Further, they show that the physiological pH of endosomes triggers lipid mixing in presents of the LASV glycoprotein by using lipid fluorophore dyes and TIRF microscopy. Interestingly, the second receptor LAMP1 increases the lipid mixing kinetics which even occurs at less acidic conditions. Thus, we have now a rational and complete view of LASV cell invasion.
All data are convincingly corroborated by several independent approaches. The experiments are clear described, and the references carefully selected. In my opinion, this work deserves an utmost praise, and I recommend this manuscript for publication.
Author Response
Lassa virus (LASV) surface glycoprotein (GP) possesses receptor binding and fusogenic properties which are essential for the delivery of the virus genome with associated proteins into the interior of the infected target cells. It has been shown that the initial steps of LASV infection requires a receptor exchange after virus uptake into the cellular vesicles and thereafter endosomal acidification for fusion of virus membrane with the endosomal membrane. Previous reports demonstrated that this fusion process only occurs at extreme low pH (< 4), which is a non-physiological role and persisted as an unsolved enigma for several years. The authors addressed the untackled problem by using a commercial pseudovirus system based on VSV expressing the LASV glycoprotein. They demonstrated that low pH values lead to loss of virus infectivity. Infectivity and antibody binding studies at pH at 37°C or 0°C and descending pH values suggested that the GP undergoes a pH-induced transition out of the pre-fusion conformation, concurrent with the loss of function. Further, they show that the physiological pH of endosomes triggers lipid mixing in presents of the LASV glycoprotein by using lipid fluorophore dyes and TIRF microscopy. Interestingly, the second receptor LAMP1 increases the lipid mixing kinetics which even occurs at less acidic conditions. Thus, we have now a rational and complete view of LASV cell invasion.
All data are convincingly corroborated by several independent approaches. The experiments are clear described, and the references carefully selected. In my opinion, this work deserves an utmost praise, and I recommend this manuscript for publication.
We thank the Reviewer for their thorough reading and evaluation of the manuscript. We are encouraged by their enthusiasm for our work.
Reviewer 2 Report
Bulow U. et al. studied the pH-dependent triggering of the spike complex from the Lassa virus. Using pseudoviruses and TIRF microscopy he shows that acidic pH can directly induce membrane fusion and that the binding to LAMP1 potentiates this triggering. Also, incubation of the spike at low pH inactivates the cell entry of pseudoviruses. This study corroborates insights that were originally made by Cohen-Dvashi H. et al., and was further studied by Hulseberg C.E. et al.
Albeit the conclusions of this study are not novel by themselves, this study examines the triggering of the Lassa virus spike complex using experimental tools that were not used before and hence warrant publication.
Addressing the comments below may improve the manuscript.
The authors refer to a potential intermediate conformation of the spike, which was also observed by Li S. et al., and was suggested by Cohen-Dvashi H. et al., as a state that could explain the kinetics that they observe. Using their current setup, the authors may actually explore the formation of this intermediate state. Specifically, pH-inactivated pseudoviruses that cannot enter cells may bear spikes that are trapped at this irreversible intermediate conformation. The authors could use TIRF microscopy to see if lipid mixing can be observed on a lipid surface that presents immobilized LAMP1 when pH-inactivated pseudoviruses are used. Quantifying the abundance of fusion events before and after such pH-inactivation and comparing that to the reduction in cell entry should provide a good answer.
No statistical analyses are shown in any of the figures. It will be important to state which of the changes in figures 1, 2, and 4 are indeed statistically significant. If for some reason it is impossible to perform statistical analysis for a particular set of experimental data, the authors should explain why.
Author Response
Bulow U. et al. studied the pH-dependent triggering of the spike complex from the Lassa virus. Using pseudoviruses and TIRF microscopy he shows that acidic pH can directly induce membrane fusion and that the binding to LAMP1 potentiates this triggering. Also, incubation of the spike at low pH inactivates the cell entry of pseudoviruses. This study corroborates insights that were originally made by Cohen-Dvashi H. et al., and was further studied by Hulseberg C.E. et al.
Albeit the conclusions of this study are not novel by themselves, this study examines the triggering of the Lassa virus spike complex using experimental tools that were not used before and hence warrant publication.
Addressing the comments below may improve the manuscript.
The authors refer to a potential intermediate conformation of the spike, which was also observed by Li S. et al., and was suggested by Cohen-Dvashi H. et al., as a state that could explain the kinetics that they observe. Using their current setup, the authors may actually explore the formation of this intermediate state. Specifically, pH-inactivated pseudoviruses that cannot enter cells may bear spikes that are trapped at this irreversible intermediate conformation. The authors could use TIRF microscopy to see if lipid mixing can be observed on a lipid surface that presents immobilized LAMP1 when pH-inactivated pseudoviruses are used. Quantifying the abundance of fusion events before and after such pH-inactivation and comparing that to the reduction in cell entry should provide a good answer.
We thank the Reviewer for their insight and suggestion. Indeed, as the reviewer suggests, we performed the lipid mixing assay using immobilized LAMP1 and pH-inactivated pseudovirions. No detectable dequenching events were observed, leading us to believe that the presence of LAMP1 is not sufficient to rescue pH-inactivated pseudovirus. This is now noted on line 139 and on line 169 of the text.
No statistical analyses are shown in any of the figures. It will be important to state which of the changes in figures 1, 2, and 4 are indeed statistically significant. If for some reason it is impossible to perform statistical analysis for a particular set of experimental data, the authors should explain why.
Statistical analyses have been clarified and added to the manuscript. In Figures 1 and 2, one-way ANOVA was used to determine significant differences across groups, and p-values for multiple individual comparisons between experimental and standard (pH 7) are denoted by asterisks. In Figure 4, error bars reflect the 95% confidence intervals determined in the fitting. Asterisks have been included to indicate significance (p < 0.05).
Reviewer 3 Report
This brief report makes nice use of pseudotyped virions and single-particle microscopy to characterize in detail important aspects of Lassa virus GPC-mediated membrane fusion. The authors demonstrate that both physiological temperature and endosomal pH are required for triggering GPC on pseudotyped VSV particles, thereby inactivating subsequent infectivity. While not highlighted in the text, this work adds to evidence debunking published reports that nonphysiological acidic pH (pH 3.5) may be necessary for fusion. They confirm that the pH-induced loss of infectivity is due to conformational changes in GPC by using several neutralizing monoclonal antibodies specific to the prefusion state. Interestingly, the binding of several non-neutralizing antibodies is unaffected by exposure to acidic pH. The membrane-fusion process is further investigated by using single-particle microscopy to detect and quantify the rate at which virion particles (labeled with quenching amounts of DiD) fuse to planar lipid membranes (displaying a fluorescent pH sensor). By co-expressing a fusion-defective HA0 on the pseudotyped particles and incorporating sialic-acid bearing lipids in the planar membrane, they are cleverly able to examine GPC fusion without the need for (and contributions of) the virus’ cell-surface receptor a-dystroglycan. This experimental set-up makes clear that binding to the physiological receptor is likely important for adhesion but not specifically for conformational changes leading to fusion. By quantitating the timing of DiD dequenching, they were able to show that the rate of membrane fusion increases as pH is lowered to nominal endosomal pH and by pre-incubation with the reported endosomal co-receptor LAMP1. Analysis of the gamma distribution of time-to-fusion among particles suggests ~4 rate-limiting steps prior to fusion, and this estimate is invariant regardless of pH or binding to soluble LAMP1. This finding may reflect a series of structural intermediates or the requirement to recruit addition GPC trimers. Together, the present studies reinforce the requirement for physiological pH and temperature for fusion activation, the lack of a requirement for a specific cell-surface receptor, and the supportive (but not required) role of LAMP1 co-receptor. While not necessarily breaking new ground in our understanding of GPC membrane fusion, the study does provide elegant and independent evidence in this regard. The manuscript is clear and well written.
Minor comment:
- Fig 3C – the vertical gray box at times ≤0 intersects the pH-sensing fluorescence signal at its midpoint. While this point is reasonably used to determine time-to-fusion, the labeling in the horizontal gray box (pH 7.5) is confusing – the initial drop in fluorescence would seem to occur under continuing pH 7.5. Is the fluid pH changed at the x-axis origin? at the initial drop in fluorescence? Might simply remove gray boxes and leave the exact timing for buffer changes unspecified. The fluorescence trace and the 50% t=0 provides all the information needed to assess the results.
Author Response
This brief report makes nice use of pseudotyped virions and single-particle microscopy to characterize in detail important aspects of Lassa virus GPC-mediated membrane fusion. The authors demonstrate that both physiological temperature and endosomal pH are required for triggering GPC on pseudotyped VSV particles, thereby inactivating subsequent infectivity. While not highlighted in the text, this work adds to evidence debunking published reports that nonphysiological acidic pH (pH 3.5) may be necessary for fusion. They confirm that the pH-induced loss of infectivity is due to conformational changes in GPC by using several neutralizing monoclonal antibodies specific to the prefusion state. Interestingly, the binding of several non-neutralizing antibodies is unaffected by exposure to acidic pH. The membrane-fusion process is further investigated by using single-particle microscopy to detect and quantify the rate at which virion particles (labeled with quenching amounts of DiD) fuse to planar lipid membranes (displaying a fluorescent pH sensor). By co-expressing a fusion-defective HA0 on the pseudotyped particles and incorporating sialic-acid bearing lipids in the planar membrane, they are cleverly able to examine GPC fusion without the need for (and contributions of) the virus’ cell-surface receptor a-dystroglycan. This experimental set-up makes clear that binding to the physiological receptor is likely important for adhesion but not specifically for conformational changes leading to fusion. By quantitating the timing of DiD dequenching, they were able to show that the rate of membrane fusion increases as pH is lowered to nominal endosomal pH and by pre-incubation with the reported endosomal co-receptor LAMP1. Analysis of the gamma distribution of time-to-fusion among particles suggests ~4 rate-limiting steps prior to fusion, and this estimate is invariant regardless of pH or binding to soluble LAMP1. This finding may reflect a series of structural intermediates or the requirement to recruit addition GPC trimers. Together, the present studies reinforce the requirement for physiological pH and temperature for fusion activation, the lack of a requirement for a specific cell-surface receptor, and the supportive (but not required) role of LAMP1 co-receptor. While not necessarily breaking new ground in our understanding of GPC membrane fusion, the study does provide elegant and independent evidence in this regard. The manuscript is clear and well written.
We thank the Reviewer for their support and encouragement.
Minor comment:
- Fig 3C – the vertical gray box at times ≤0 intersects the pH-sensing fluorescence signal at its midpoint. While this point is reasonably used to determine time-to-fusion, the labeling in the horizontal gray box (pH 7.5) is confusing – the initial drop in fluorescence would seem to occur under continuing pH 7.5. Is the fluid pH changed at the x-axis origin? at the initial drop in fluorescence? Might simply remove gray boxes and leave the exact timing for buffer changes unspecified. The fluorescence trace and the 50% t=0 provides all the information needed to assess the results.
Indeed the Reviewer correctly points out the misleading nature of the color gradient in Figure 3. The loss of 6-carboxyfluorescein fluorescence does indeed occur at acidic pH, not at neutral pH as suggested by the schematic in Figure 3C. As suggested, we have removed the grey box.